# Understanding the Impact of the COVID-19 Pandemic on Mental Health among a Sample of University Workers in the United Arab Emirates

**DOI:** 10.3390/healthcare12111153

**Published:** 2024-06-06

**Authors:** Anamika V. Misra, Heba M. Mamdouh, Anita Dani, Vivienne Mitchell, Hamid Y. Hussain, Gamal M. Ibrahim, Reham Kotb, Wafa K. Alnakhi

**Affiliations:** 1Department of Health Sciences—Social Work Program, Higher Colleges of Technology, Sharjah P.O. Box 7946, United Arab Emirates; amisra@hct.ac.ae (A.V.M.); anitadani@gmail.com (A.D.);; 2Department of Data Analysis, Research and Studies, Dubai Health Authority, Dubai P.O. Box 4545, United Arab Emiratesgmibrahim@dha.gov.ae (G.M.I.); 3Department of Family Health, High Institute of Public Health, Alexandria University, Alexandria 5424041, Egypt; 4Department of Public Health, College of Health Sciences, Abu Dhabi University, Abu Dhabi P.O. Box 59911, United Arab Emirates; rehamkotb@gmail.com; 5Primary Health Care Department, High Institute of Public Health, Alexandria University, Alexandria 5424041, Egypt; 6Department of Family and Community Medicine and Behavioural Sciences, College of Medicine, University of Sharjah, Sharjah P.O. Box 26666, United Arab Emirates; walnakhi@sharjah.ac.ae

**Keywords:** mental health, anxiety, depression, resilience, COVID-19 pandemic, United Arab Emirates, university workers

## Abstract

Research on the mental health of university staff during the COVID-19 pandemic has uncovered a high prevalence of probable anxiety, depression, and post-traumatic stress disorder among academic and non-academic staff in many parts of the world. This study aimed to assess the prevalence of anxiety, depressive symptoms, and resilience among a sample of faculty and staff members working in the Higher College of Technology campuses in the UAE. From September to November 2021, a cross-sectional study was carried out using an online survey. The Generalized Anxiety Disorder 7-item scale, The Patient Health Questionnaire (9-items), and the Connor–Davidson Resilience Scale were used to assess anxiety, depression, and resilience. The impact of COVID-19 was assessed using a designated list of questions. The results demonstrated that the COVID-19 pandemic had impacted the mental health of the studied sample of university workers, with almost 16% of the participants having moderate-to-severe depression and anxiety symptoms. This study highlighted significant differences in the participants’ depressive and anxiety symptoms due to sociodemographic differences. Depression and anxiety symptoms were most prevalent among females, those of UAE nationality, and never-married workers, with females scoring 5.81 on the PHQ-9 compared to only 4.10 in males, *p* = 0.004 *. UAE-national participants had significantly higher mean PHQ-9 scores than their non-national counterparts (6.37 ± 5.49 SD versus 4.77 ± 5.1 SD, respectively, *p* = 0.040 *). Overall, the total mean scores of all participants were below the assumed cut-off threshold of having a high resilience level (29.51 ± 7.53 SD). The results showed a significant difference in severe depression symptoms as a result of the impact of COVID-19. These results could imply that the COVID-19 pandemic might have augmented negative mental health impacts on this sample of university workers. This study highlighted some areas where the responsible authorities can intervene to further protect and enhance the mental health of university workers, particularly after the COVID-19 pandemic.

## 1. Introduction

The emergence of the novel coronavirus, COVID-19, in late 2019 marked the onset of a global health crisis that has left an indelible bio-psycho-social mark on societies and individuals worldwide [1]. While the virus posed a significant threat to physical health, the pandemic itself cast a long shadow over mental well-being, affecting diverse population groups [2,3,4,5]. Life challenges, such as the COVID-19 pandemic, along with stress, can trigger common mental health problems, such as anxiety and depression, which may require proper coping strategies in order to maintain individual mental well-being [6].

The COVID-19 crisis has presented unparalleled challenges for organizations, necessitating innovative solutions to safeguard both operations and employee well-being [7]. Human resource managers are tasked with developing intervention plans that ensure efficient and continuous communication with employees, considering the ever-evolving situation [7]. Mental health issues in the workplace have become a normalized aspect of life, prompting employers to increasingly recognize the importance of mental health support [8]. Furthermore, a comparative analysis of mental health, stigma, and work culture in the workplaces before and during the pandemic in the United States revealed a concerning trend, with the prevalence of mental health conditions on the rise from the years 2019 to 2021, particularly among younger workers [9]. These challenges have driven more employees to leave their jobs for mental health reasons, highlighting the need for robust support systems and an empathetic work culture [9].

Research on the mental health of university staff during the COVID-19 pandemic has uncovered a high prevalence of probable anxiety, depression, and post-traumatic stress disorder among academic and non-academic staff at higher educational institutes [2,10,11]. Since the commencement of the pandemic, universities and other educational institutions have experienced unprecedented challenges due to online education, uncertainties related to the courses’ evaluation, and learning outcome assessment [12]. University staff in particular have confronted several obstacles due to remote teaching, remote-teaching-linked technical problems, the lack of face-to-face interactions, and many other issues [13,14]. These obstacles, along with the other daily life difficulties that were triggered by the pandemic, have negatively affected the mental health of teachers and university staff around the globe [11,12].

The United Arab Emirates (UAE), a diverse and rapidly developing nation, hosts a substantial international academic workforce and a wide range of non-academic personnel within its universities [15]. As the UAE’s higher education sector plays a pivotal role in national development and global integration, understanding the mental health status of university workers is of paramount importance. It not only directly impacts the well-being of this workforce but also has implications for the quality of education, research, and administrative services provided by these institutions [16]. Additionally, the government has shown a commitment to fostering a culture of well-being and inclusivity, making it crucial to inform evidence-based policies and interventions tailored to the specific needs of university workers [17].

Despite the growing body of literature on the mental health repercussions of the pandemic, there is a noticeable scarcity of research focusing specifically on university workers in the UAE [2,3,18,19,20]. While studies conducted in other countries have provided valuable insights into the experiences of faculty and staff during these challenging times, the unique sociocultural context of the UAE necessitates a dedicated examination of this population [10,11,13,21]. To address this gap, our research aims to investigate the prevalence of anxiety and depressive symptoms within a specific population group—faculty and staff members at the Higher Colleges of Technology, one of the higher education institutions in the UAE. Furthermore, we endeavor to assess the resilience of university staff and gauge the extent to which the COVID-19 pandemic has influenced their mental health [22].

## 2. Materials and Methods

### 2.1. Study Design and Sample

A cross-sectional study was conducted among a sample of university staff and faculty members who were working at the HCT campuses across the UAE. This study used a structured self-administered questionnaire for data collection. Participants were recruited via announcements through the email network of the HCT educational institution [20]. An inconvenience sampling technique was followed in this study in which the selection of participants from the target population was based on ease of access.

Online data collection took place from September to November (2021). The responses were extracted using an electronic survey via the Google survey tool (Google Forms). The electronic survey was conducted in accordance with the Checklist for Reporting Results of Internet E-Surveys [23]. Participants were asked for consent approval before participation.

No exclusion criteria were followed. The sample included those who were working at the university campuses (part-time or full-time) and voluntarily agreed to participate in the online survey. The mean completion time for the survey was 12 min. Based on the Raosoft sample size estimation, the minimum required sample for this study was 281, with a confidence interval of 95.0 and a 0.5 margin of error [24]. Out of the total surveys sent, 317 staff voluntarily responded with a response rate of 14.9%. Only 313 participating staff fully completed the survey and were analyzed for this study.

### 2.2. Variables and Measures

In addition to questions regarding COVID-19-related items, the survey comprised measures regarding socio-demographics and the current depression, anxiety, and resilience state of the participants. Socio-demographics included gender, age groups, nationality, marital status, employment status (whether faculty teaching staff or non-teaching administrative staff), and Emirate of residence within the UAE. Nationality was dichotomized into UAE nationals and non-UAE nationals. Marital status was grouped into ever-married, which included married and divorced/widowed, and single or never-married participants.

The impact of COVID-19 on the participants was assessed using an outcome variable named “the COVID-19 impact”. The variable was dichotomized into “impacted by COVID-19” or “not impacted by COVID-19”. Seven questions assessed if the respondents were impacted by COVID-19 in some way or another. “Impacted by COVID-19” was defined if the participants answered “yes”, they were diagnosed with COVID-19 themselves or a close family or friend was (3 questions), witnessed a COVID-19-related death of a close friend or a family member (3 questions), or had a high exposure to COVID-19 at the workplace in the year preceding the survey (2 questions). Exposure at the workplace included settings where they could have had a high risk of COVID-19 exposure. The respondents who answered no to all of the seven questions were grouped in the category of “not impacted by COVID-19”.

### 2.3. The Scales Used to Assess the Mental Health Status

(I)The Patient Health 9-Item Questionnaire (PHQ-9)

The PHQ-9 is a brief 9-item depression assessment measure adopted from the full PHQ. The PHQ-9 has been previously recognized as a valid and reliable instrument in both clinical and non-clinical samples [25,26]. Participants were asked to rate their depression over the past 2 weeks using a 4-point Likert scale ranging from 0 (not at all) to 3 (nearly every day). Total scores, obtained by summing the responses to each item, range from 0 to 27. We adopted PHQ-9 cut-off scores of ≤9 and ≥10 that suggest minimal-to-mild depression and moderate-to-severe depression on the PHQ-9 scale, respectively [27]. The reliability of the full scale in this study was excellent (Cronbach alpha = 0.87).

(II)The Generalized Anxiety Disorder 7-Item Questionnaire (GAD-7)

The Gad-7 is a widely used self-reporting scale to assess the symptoms of anxiety. It consists of 7 items that measure anxiety over the past 2 weeks. Items are rated on a 4-point Likert scale ranging from 0 (not at all) to 3 (nearly every day). The GAD-7 score is calculated by assigning scores of 0, 1, 2, and 3 and then adding together the scores for the seven questions, adding up to a maximum of 21 total scores. A threshold of 10 is considered to categorize minimal-to-mild and moderate-to-severe levels of anxiety on the GAD-7 [28,29]. The reliability of the scale among the current sample was excellent (Cronbach alpha = 0.91).

The variables between the three research instruments had an excellent correlation score, with a correlation coefficient (r) of 0.91. In addition, the reliability of the variables between the three scales was high (Cronbach alpha = 0.83).

(III)Connor–Davidson Resilience Scale (CD-RISC-10)

The CD-RISC-10 was developed to provide a valid and reliable measurement of resilience and to establish reference values for being resilient (their ability to tolerate and overcome adverse situations such as illness, pressure, and failure) or non-resilient in the general population and in clinical trials [30,31]. It consists of 10 items; each item is rated on a 5-point Likert scale (0 = not true at all, 1 = rarely true, 2 = sometimes true, 3 = often true, and 4 = true nearly all of the time). A higher total score indicates greater resilience. Due to the lack of a recognized cut-off point, resilience scores were categorized into high resilience (score ≥ 33) and normal or low resilience (score ≤ 32) [32]. The reliability of the scale among the present sample was excellent (Cronbach alpha = 0.89).

The prevalence (estimates) of depression and anxiety were determined using cut-off points based on PHQ-9 and GAD-7 scale validation [26,28]. In this study, depression was defined as a total score of ≥10 on the PHQ-9 scale, indicating a case of moderate-to-severe depression. Anxiety was defined using the GAD-7 instrument with a total score of ≥10, indicating a case of moderate-to-severe anxiety. The prevalence of depression or anxiety was estimated by dividing the number of staff members who exceeded the cut-off score by the total number of participating staff. Cut-off scores were considered at 5, 10, and 15 points for mild, moderate, and severe anxiety, respectively.

### 2.4. Statistical Analysis

Data coding, cleaning, and analysis have been carried out using IBM SPSS (Version 22.0, IBM SPSS, IBM Corp., Armonk, NY, USA). Cronbach’s alpha coefficients were calculated to indicate scale reliability. Descriptive statistics, including means, standard deviations (±SD), and percentages, were used to illustrate participant demographics. The normal distribution of data was verified using box plots and histograms. A complete case analysis was considered in this study, with 4 missing cases excluded from the statistical analysis. The equality of variances was checked using Levene’s test. After checking the sample distribution, the current sample was normally distributed; accordingly, we used parametric tests. An independent sample *t*-test was used to compare the mean scores between the category for participants impacted by COVID-19 and the mean scores of each of the three scales (depression, anxiety, and resilience). The mean scores of participants’ anxiety, depression, and resilience were compared with demographic characteristics using an independent-sample *t*-test, one-way analysis of variance (ANOVA). An independent samples *t*-test was used to compare the mean scores of the three psychometric scales (anxiety, depression, and resilience scales) between different socio-demographic groups and between the COVID-19 impact categories, separately. A univariate analysis of variance (ANOVA) was used to examine if the mean scores of the three psychometric scales (anxiety, depression, and resilience scales) were different between the COVID-19 impact category and participants’ gender. A statistical significance of ≤0.05 was considered in this study, with 95% confidence intervals.

### 2.5. Ethical Approval and Consent

This study was approved by the Higher College of Technology Research Ethics Review Board Committee. Participants gave online written consent to participate in the study prior to starting the survey.

## 3. Results

The socio-demographic characteristics of the participants are shown in Table 1. It reveals that 55.5% of the staff were females, and the majority (82.1%) fell under the ever-married category. Most of the participants (82.4%) were non-UAE nationals. Participants’ age ranged from 20 to 62 years, with the highest proportion in the age group of 34–49 years (48.9%). Overall, 71.9% of the sample were working as faculty and teaching staff, while the remaining were working in administrative jobs. As for the Emirate of residence, a similar proportion (39%) reported living in the Emirate of Dubai or Northern Emirates.

Figure 1 illustrates the distribution of the participants by COVID-19-related questions. Not surprisingly, the majority of the participants (85.9%) were categorized as impacted by COVID-19 (as per the COVID-19 impact questions). Less than half of the staff reported having been diagnosed with COVID-19 themselves or significant relatives/friends having been diagnosed (46.1%). Additionally, 28.4% of participants stated they knew some close relatives/friends who died from COVID-19 or its complications.

Prevalence estimates of depression, anxiety, and resilience by COVID-19 impact among the participants are shown in Table 2. Based on PHQ-9 cut-off scores (≥10), the self-reported prevalence of moderate-to-severe depression symptoms was 15.7%, and it was significantly higher in participants who were categorized as impacted by COVID-19 than those who were not impacted (17.5% and 4.50%, *p* < 0.05). Moreover, taking GAD-7 cut-off scores of ≥10, the self-reported prevalence of moderate-to-severe anxiety was 15.5%. A slightly higher proportion of staff with moderate-to-severe anxiety symptoms were categorized as impacted by COVID-19 (18.9%) than those who were not impacted (15.9%). Less than one-fourth of the participants (37.3%) self-reported high levels of resilience (CD-RISC-10 score ≥ 33); among them, those who were categorized as not impacted by COVID-19 (45.5%) had a higher level of resilience than their impacted counterparts (36.1%). Prevalence estimates of depression, anxiety, and resilience as measured by PHQ-9, GAD-7 and CD-RISC-10 cut-off scores by gender among the participants are illustrated in Figure 2. Overall, female participants had more frequent moderate and severe depression and anxiety (19.7% and 22.7%, respectively) than their male counterparts (10.7% and 12.9%, respectively). An equal proportion of females and males had higher resilience scores (37.2% and 37.1%, respectively).

The independent sample *t*-test used for comparison between the mean scores (±SD) of the psychometric scales by COVID-19 impact is presented in Table 3. Notably, the total mean scores (±SD) of all participants on the three psychometric scales used were below the assumed cut-off threshold of depression (5.04 ± 5.19), anxiety (5.13 ± 5.58), or high resilience (29.51 ± 7.53). Significantly higher mean PHQ-9 (±SD) scores were found among those who reported to be impacted by COVID-19 (5.37 ± 5.30) than those who were non-impacted (3.05 ± 3.94). No statistically significant differences were detected in the mean GAD-7 or CD-RISC-10 scores for those who were impacted by COVID-19 and those who were non-impacted.

Table 4 shows the comparison of the mean scores (±SD) of the three psychometric scales by sociodemographic characteristics for the participants using an independent sample *t*-test. Regarding the PHQ-9, there was a statistically significant difference between males (4.10 ± 4.89) and females (5.81 ± 5.31), *p* = 0.004 *. Similarly, the mean scores of the GAD-7 were significantly different between male (4.14 ± 5.29) and female (5.94 ± 5.69) participants, *p* = 0.004 *. UAE-national participants had significantly higher mean scores (±SD) for PHQ-9 than their non-national counterparts (6.37 ± 5.49 versus 4.77 ± 5.10, respectively, *p* = 0.040 *). Similarly, the mean anxiety score ± SD was higher in UAE nationals than non-nationals (7.39 ± 6.03 versus 4.65 ± 5.38, respectively; *p* < 0.001 *). As for the employment status, both those who were working as teaching staff or non-teaching administrative staff had quite similar PHQ-9, GAD-7, and CD-RISC-10 scores. Regarding the marital status, single/never-married participants had significantly lower GAD-7 scores and were significantly less resilient (higher CD-RISC-10) than their ever-married counterparts (6.50 ± 5.97 versus 4.83 ± 5.45, *p* = 0.032 * for GAD-7) and (27.20 ± 9.14 versus 30.01 ± 7.05, *p* = 0.011 * for CD-RISC-10).

Table 5 reveals the interaction between the effects of the COVID-19 impact and gender on the mean scores of the three psychometric scales using a two-way ANOVA test. There was a statistically significant interaction between the effects of gender and the COVID-19 impact on depression scores. In particular, male participants who were categorized as impacted by COVID-19 had a slightly significant higher mean PHQ-9 score than those who were not impacted (4.38 ± 5.11 versus 2.80 ± 3.65, respectively; *p* = 0.015). Similarly, females who were categorized as impacted by COVID-19 (interaction term) had a significantly higher mean PHQ-9 score than those who were not impacted (6.09 ±5.36 versus 3.50 ± 4.46, respectively; *p* = 0.015). No statistically significant differences were detected in the mean scores of anxiety and resilience for the participants regarding COVID-19 impact.

We tested for interaction between the COVID-19 impact variable and other socio-demographics and the mean scores of the PHQ-9, GAD-7, and CD-RISC-10 psychometric scales using the tests of between-subjects effects; unfortunately, the results were non-statistically significant.

## 4. Discussion

### 4.1. Depression and Anxiety Symptoms and the Impact of COVID-19

Globally, studies conducted during the COVID-19 pandemic show a significant impact on mental health among various population groups [33]. This current cross-sectional study is based on a sample of university-level and academic institution staff across HCT campuses in the UAE. This study examined the prevalence of depression, anxiety, and resilience during the COVID-19 pandemic and evaluated the potential association between these psychometric scores and specific sociodemographic characteristics. The results of this study demonstrated that the COVID-19 pandemic had impacted the mental health of the studied sample of university staff in some capacity, with almost 16% of the participating staff reporting moderate-to-severe depression and anxiety symptoms. These levels were most prevalent among females, those who were UAE nationals, and never-married participants.

The findings from this study indicate that the levels of depression and anxiety were lower than what was reported for the general public in the UAE during the COVID-19 pandemic, where 32.8% and 26.4% of the participants reported depression and anxiety symptoms, respectively [34]. These findings may be attributed to the possible high level of knowledge about the COVID-19 disease among academic staff and university-level workers. The level of depression and anxiety reported in this study was less than what was reported among the university staff in various settings during the pandemic [14,21,35,36]. It is worth mentioning that the prevalence estimates of depression or anxiety greatly varied across studies depending on the severity and the instruments used [37]. It is recommended to consider COVID-19 and other crisis-related changes in workload models, as well as in working practice planning. Some workplace factors that might negatively impact and add another layer of mental health issues include emotional drain, the place, and work–life balance, and these could apply to universities’ workplaces [9].

This study was conducted during the COVID-19 pandemic; accordingly, higher mean PHQ-9 and GAD-7 scores were reported among the participants who were categorized as impacted by COVID-19 than those who were not. These results could imply that the COVID-19 pandemic might have augmented the negative mental health impact on this sample of university-level workers. Our findings corroborate the global literature that underscores the significant psychological repercussions of the pandemic on various populations [19,31,38,39,40]. A study argued that stressful situations might have a greater impact on individuals with higher educational backgrounds [11,41]. To our knowledge, no published research evaluated the depression and anxiety symptoms at the university level or among workers in the UAE before the pandemic. For this reason, we are unable to compare the prevalence estimates of depression and anxiety with pre-pandemic ones among the existing sample.

### 4.2. Socio-Demographic Differences and the Participants’ Depression and Anxiety Symptoms

In response to the pandemic, numerous studies have explored the substantial impact of COVID-19 on mental health, shedding light on the challenges faced by various demographic groups in coping with the associated stressors and uncertainties [11,15,16,36].

The prevalence of depression and anxiety in this existing study is evident in both genders but more pronounced among females. This gender difference supports many previous studies in which depression and anxiety scores were higher in women compared to men [36,42,43]. The possible explanation for such an outcome can be linked to the wider emotional threshold and sensitivity tolerance women are characterized with, compared to men [44,45]. Prior research suggests that women might be more susceptible to psychological distress during pandemics due to a combination of biological, psychological, and socio-cultural factors [46].

Interestingly, this study found that UAE nationals had higher mean scores for depression and anxiety in relation to COVID-19 exposure than their non-national counterparts in the same workplace. There is conflicting evidence about the relationship between depression and ethnicity [47,48,49,50]. It is plausible that UAE nationals had different stressors or perceived the pandemic differently due to cultural or societal reasons. Evidence of the measurement invariance of the PHQ-9 scale regarding ethnicity was reported by some researchers, suggesting that the observed differences in depressive symptoms may not only be attributed to the ethnicity factor [51]. This area would benefit from a further in-depth exploration of different population groups in the UAE.

Unexpectedly, in this present sample, single or never-married participants had significantly lower anxiety scores than their married counterparts. This is not in accordance with the work of many authors. According to the results of a meta-analysis conducted in India, the prevalence of anxiety symptoms was higher in females and in unmarried individuals [52]. Usually, unmarried people are of a younger age and might have less experience and fewer stress-coping techniques, which contradicts our current findings [33,41]. We hypothesize that being single or unmarried in this present sample meant fewer financial and other family-related responsibilities, which are expected to be related to the COVID-19 pandemic.

### 4.3. Levels of Resilience and the Impact of COVID-19

Resilience has been highlighted as a crucial protective factor during times of crisis [53]. There are a few research studies that looked into how resilience changed throughout the COVID-19 epidemic, and, consistent with our current findings, they found that resilience scores did not change significantly over time during the pandemic [54,55,56]. This finding could be explained in the context of the fact that university-level workers or academic staff somehow have high or normal levels of resilience compared to other population groups. It is noteworthy to mention that those who were not impacted by COVID-19 showcased greater resilience. This might suggest that individuals with higher resilience were less likely to perceive themselves as being affected by the pandemic and manage its implications more effectively.

This current study is significant, as it surveyed university workers’ and academic staff’s resilience while they were working and dealing with everyday stressors and life during the COVID-19 pandemic. Notably, more than one-third of the participants self-reported high levels of resilience (CD-RISC-10 score of ≥32), compared to 11.6% reported in university students in a similar setting [41]. It is worth mentioning that the UAE government was effective in responding to the pandemic and providing the workers in the education system—and residents at large—with the needed support, which therefore helped to decrease the impact [57,58]. Not surprisingly, single/never-married participants in our sample scored less in resilience than their never-married counterparts. This was supported by the findings of Peng J. and his colleagues, who reported marriage to have been a protective factor for well-being and resilience during the COVID-19 pandemic [59].

This study’s finding that single/never-married participants exhibited significantly lower resilience compared to their ever-married counterparts aligns with the existing literature, which suggests that having close interpersonal relationships, such as marriage, can act as a buffer against psychological distress.

### 4.4. Strengths and Limitations

This current study has some strengths. This study adds to the body of knowledge on how the pandemic affected the mental health and well-being of different population groups, particularly among a sample of university-level workers in the UAE. This current study’s utilization of recognized psychometric scales provides validity to its findings. The reliability of these scales in this specific sample is commendable, with all Cronbach’s alpha values indicating excellent internal consistency. Furthermore, testing the demographic factors also enabled us to report on the groups that seem to experience a greater mental health burden and to propose a role for future interventions.

Still, this study must report some limitations. As in similar cross-sectional self-reported surveys that look into mental health topics, causality cannot be determined, and the results could not be generalizable because they only reflect the opinions of a small subset of UAE university workers. Moreover, the sample primarily consisted of university staff, which might not represent the wider UAE population. Furthermore, the low response rate, albeit typical of online surveys, suggests potential non-response bias. There might have been a meaningful distinction between the participants who decided to take part in the study and those who did not. Yet, neither of these elements should have an impact on the respondents’ answers, as the data were adequately managed to address the true values and impacts of the measured variables.

## 5. Conclusions

According to these results, the mental health of the studied group of university-level workers and university staff has been influenced by the COVID-19 pandemic in terms of depressive symptoms. Prevalence estimates show that moderate-to-severe depression and anxiety symptoms were self-reported by a considerable percentage of the sample (as per PHQ-9 and GAD-7 cut-off scores). The COVID-19 pandemic was associated with an increase in the symptoms of anxiety and depression in this cohort. Additionally, cross-analysis with the participants’ demographic characteristics showed how the current sample’s mental health was influenced by differences in gender, marital status, and nationality, suggesting a need for future targeted interventions. Moreover, four in ten of the university staff sampled in this study reported high levels of resilience; nevertheless, the influence of COVID-19 on CD-RISC-10 scores was not statistically significant. Some variations in the mean resilience scores varied significantly by marital status and nationality grouping.

Therefore, our findings highlight some areas where the responsible authorities can intervene to further protect and enhance the mental health of university workers after the COVID-19 pandemic through focusing on the most affected groups (females, married individuals, and certain nationalities). Ensuring that university workers have no barriers to accessing needed mental healthcare is vital. By scrutinizing this particular group, our research sheds light on the ways in which the pandemic’s mental health impact varied across different subsets of the population. Virtual consultations or hotlines could be introduced to ensure staff confidentiality and privacy. Regular monitoring of the psychological status of university workers by certified mental healthcare providers should be guaranteed in confidential ways, particularly for vulnerable groups. Further research can include follow-ups of this sample and similar samples from various university settings to allow for an accurate assessment of the mental health status of this targeted population. Studies on the long-term psychological and behavioral effects of COVID-19 should be conducted even after the end of the pandemic.

## Figures and Tables

**Figure 1 healthcare-12-01153-f001:**
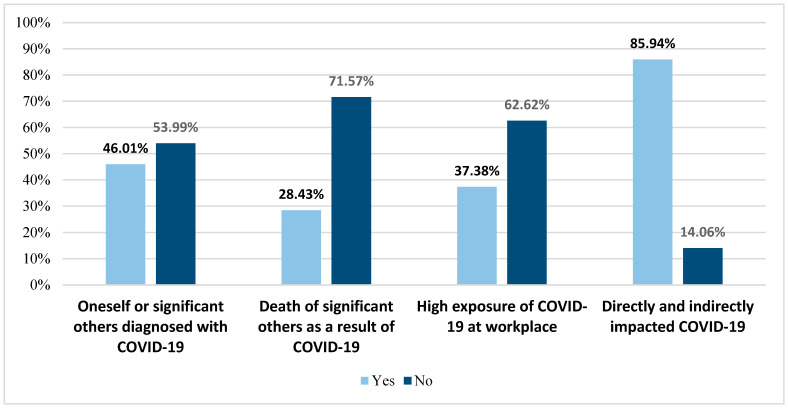
Distribution of the participants by COVID-19-related questions.

**Figure 2 healthcare-12-01153-f002:**
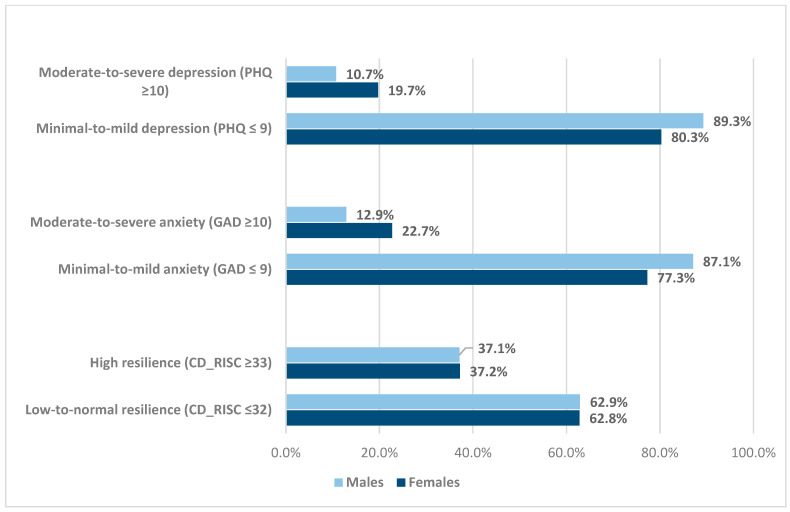
Distribution of the sample by prevalence estimates of depression, anxiety, resilience status, and gender.

**Table 1 healthcare-12-01153-t001:** Sociodemographic characteristics of the participating staff (N = 313).

	Characteristic	Number (%)
Gender	Males	138 (44.5)
	Females	172 (55.5)
Marital status	Single/Never Married	56 (17.9)
	Ever Married	257 (82.1)
Current age (in years)	20 to 33	49 (15.6)
	34 to 49	153 (48.9)
	50+	111 (35.5)
Nationality	UAE Nationals	55 (17.6)
	Non-UAE Nationals	258 (82.4)
Employment Status	Faculty and Teaching Staff	225 (71.9)
	Non-Teaching (Administrative) staff	88 (28.1)
Residence (by Emirate)	Abu Dhabi and Western Region	68 (21.7)
	Dubai	122 (39.0)
	North Emirates	123 (39.3)
Total		313 (100.0)

**Table 2 healthcare-12-01153-t002:** Prevalence estimates of depression, anxiety, and resilience (as measured by the PHQ-9, GAD-7, and CD-RISC-10 scales) among the participants by COVID-19 impact.

Psychometric Property	Impacted by COVID-19 N (%)	Not Impacted by COVID-19 N (%)	Total Sample N (%)
Depression as measured by the PHQ-9			
Minimal-to-mild depression (score ≤ 9)	222 (82.5)	42 (95.5)	264 (84.3)
Moderate-to-severe depression (score ≥ 10)	47 (17.5)	2 (4.5)	49 (15.7) *
Anxiety as measured by the GAD-7			
Minimal-to-mild anxiety (score ≤ 9)	218 (81.1)	37 (84.1)	255 (81.5)
Moderate-to-severe anxiety (Score ≥ 10)	51 (18.9)	7 (15.9)	58 (15.5)
Resilience as measured by the CD-RISC-10			
Low-to-normal resilience (≤32)	172 (63.9)	24 (54.5)	196 (62.7)
High resilience (score ≥ 33)	97 (36.1)	97 (36.1)	117 (37.3)
Total	269 (100)	44 (100)	313 (100)

* Significant at *p* < 0.05 using the Chi-square test. PHQ-9, patient health 9-item questionnaire; GAD-7, generalized anxiety disorder 7-item questionnaire; CD-RISC-10, Connor–Davidson Resilience Scale.

**Table 3 healthcare-12-01153-t003:** Independent sample *t*-test comparing the mean scores (±SD) of the PHQ-9, GAD-7, and CD-RISC-10 psychometric scales by COVID-19 impact.

Scale	Impacted by COVID-19	Not-Impacted by COVID-19	T Value	* *p* Value	Total Score
	Mean Scores (±SD)				
PHQ-9	5.37 (5.30)	3.05 (3.94)	−2.78	0.008 *	5.04 (5.19)
GAD-7	5.26 (5.59)	4.30 (5.48)	−1.06	>0.05	5.13 (5.58)
CD-RISC-10	29.58 (7.10)	29.07 (9.80)	−0.41	>0.05	29.51 (7.53)

* Significant at *p* < 0.05. PHQ-9, patient health 9-item questionnaire; GAD-7, generalized anxiety disorder 7-item questionnaire; CD-RISC-10, Connor–Davidson Resilience Scale.

**Table 4 healthcare-12-01153-t004:** Independent sample *t*-test comparing the mean scores (±SD) of the PHQ-9, GAD-7, and CD-RISC-10 psychometric scales by demographic characteristics.

Scale	Mean Scores (±SD)		T Value	* *p* Value
Characteristic	Male	Female		
PHQ-9	4.10 (4.89)	5.81 (5.31)	−2.921	0.004
GAD-7	4.14 (5.29)	5.94 (5.69)	−2.866	0.004
CD-RISC-10	29.47 (6.93)	29.53 (8.00)	−0.074	0.941
	UAE national	Non-national		
PHQ-9	6.37 (5.49)	4.77 (5.10)	2.066	0.040
GAD-7	7.39 (6.03)	4.65 (5.38)	3.320	0.001
CD-RISC-10	25.92 (8.62)	30.28 (7.07)	−3.945	0.000
	Teaching	Non-Teaching		
PHQ-9	5.17 (5.29)	4.70 (4.93)	0.519	0.474
GAD-7	5.16 (5.61)	5.05 (5.53)	0.940	0.870
CD-RISC-10	29.52 (7.82)	29.46 (6.76)	0.563	0.953
	Ever married	Single/never married		
PHQ-9	4.98 (5.29)	5.32 (4.72)	0.445	0.657
GAD-7	4.83 (5.45)	6.50 (5.97)	2.040	0.032
CD-RISC-10	30.01 (7.05)	27.20 (9.14)	−2.557	0.011

* Significant at *p* < 0.05; PHQ-9, patient health 9-item questionnaire; GAD-7, generalized anxiety disorder 7-item questionnaire; CD-RISC-10, Connor–Davidson Resilience Scale.

**Table 5 healthcare-12-01153-t005:** Two-way ANNOVA for comparing the differences in mean scores of PHQ-9, GAD-7, and CD-RISC-10 scales among the participants by COVID-19 impact and by gender.

Scale	Impacted by COVID-19	Not Impacted by COVID-19	* *p* Value
	Mean Score (±SD)		
PHQ-9			
Male	4.38 (5.11)	2.80 (3.65)	0.015
Female	6.09 (5.36)	3.50 (4.46)	
GAD-7			
Male	4.19 (5.39)	3.96 (5.10)	0.123
Female	6.07 (5.66)	4.94 (6.16)	
CD-RISC-10			
Male	29.18 (6.43)	30.60 (9.11)	0.226
Female	29.84 (7.61)	26.89 (10.83)	

* Significant at *p* < 0.05. SD, Standard Deviation; PHQ-9, patient health 9-item questionnaire; GAD-7, generalized anxiety disorder 7-item questionnaire; CD-RISC-10, Connor–Davidson Resilience Scale.

## Data Availability

The datasets generated and analyzed during this current study are not publicly available because data analysis is ongoing to study variables other than those covered in this study. The data that support the findings of this study are available upon request, but restrictions apply to the availability of these data. Data are, however, available from the authors upon reasonable request and with permission of the HCT.

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
