# Peer review of "Understanding the Impact of the COVID-19 Pandemic on Mental Health among a Sample of University Workers in the United Arab Emirates"

_healthcare, 2024, doi:10.3390/healthcare12111153_

Round 1
Reviewer 1 Report
Comments and Suggestions for Authors
Dear authors,
The manuscript is well-written and shows some potential. However, I think there are justifications and improvements required before it can be recommended for publication.
Abstract
1. Please check the format.
2. Please add a summarized background comprising the problem statement.
3. The location of the study (the scope) should be added to the methodology.
Introduction
1. It would be great if the causes of mental health issues among university workers could be summarized in a table with references. It will strengthen the motivation of the manuscript.
Materials and methods
1. The authors should inform the population about the study before justifying the minimum number of participants.
2. The design of the research instruments should be linked with the introduction to justify the questions being formed.
3. The statistical analysis is well-written. However, please justify the selection of the analysis techniques; the authors can also relate to the data distribution.
Results
1. The results presented are fine. However, I think that they are very fundamental. Can a higher impact analysis be implemented, such as the relationship between demography/causes and mental health perceptions?
2. Are the variables between the three research instruments being correlated?
Discussions
1. Discussions are well-written, but 4.2 and 4.3 will be more impactful with a better statistical analysis.
2. 4.4 should be under conclusion.
Thank you.
Author Response
We thank the editorial team and the reviewers for their careful evaluation of the manuscript, which has helped us to substantially improve our manuscript, titled “Understanding the Impact of COVID-19 on Mental Health among a sample of University Workers in the United Arab Emirates.” We have provided a point by point response for each suggestion and question raised by reviewer 1 in the text below. We hope our responses are clear and satisfactory.
.
Reviewer’s comment |
Feedback |
Abstract |
|
1. Please check the format. 2. Please add a summarized background comprising the problem statement.
3. The location of the study (the scope) should be added to the methodology. |
|
Introduction |
|
The design of the research instruments should be linked with the introduction to justify the questions being formed.
|
|
Methodology |
|
1. The authors should inform the population about the study before justifying the minimum number of participants.
2. The statistical analysis is well-written. However, please justify the selection of the analysis techniques; the authors can also relate to the data distribution.
|
2. As per the reviewer’s feedback, the selection of analytical testes was mentioned in the methodology section in lines 175-179 by adding “After checking the sample distribution, the current sample was normally distributed, accordingly we used”. And modifying the other sentences. |
Results |
|
1. The results presented are fine. However, I think that they are very fundamental. Can a higher impact analysis be implemented, such as the relationship between demography/causes and mental health perceptions?
2. Are the variables between the three research instruments being correlated?
|
Response: 1. Kindly note that, we actually ran some other tests to assess the impact COVID-19 on the socio-demographics. The Independent-sample t tests were used to test the difference of the mean scores for the three scales and the COVID-19 impact. Only for female participants the difference in mean scores for PHQ-9 and GAD-7 those impacted by COVID-19 and not impacted by COVID-19. We also tested for the interaction between the COVID-19 impact and other socio-demographics and the mean scores of PHQ 9, GAD-7, and CD-RISC 10 psychometric scales using the tests of Between-Subjects Effects and unfortunately all gave non-statistically significant results. Mostly due to the small sample size and the smaller sample of those who had a COVID-19 impact, a lot of variables gave insignificant results. We can provide the data in appendixes, though no significant results were obtained (however, it will be of no benefit).
2- As suggested, correlation coefficient for each scale was measured and shown in the text in lines 151- 153. “The variables between the three research instruments had an excellent correlation, with a correlation coefficient of 0.91. In addition, the reliability of the variables between the three scales was high (Cronbach alpha = 0.83).“ |
Discussion |
|
1. Discussions are well-written, but 4.2 and 4.3 will be more impactful with a better statistical analysis.
2. 4.4 should be under conclusion
|
1- As suggested by the reviewer, association between the socio-demographic differentials and the participants’ depressive and anxiety symptoms and the impact of COVID-19 needed a better statistical analysis. kindly note that we tried some parametric and non-parametric tests with no further significant associations revealed between these variables. Section 4.4 (Strengths and limitations) came under the discussion as per the healthcare Journal guidelines and many other journals that requested its position to be under discussion.
|
Reviewer 2 Report
Comments and Suggestions for Authors
Dear authors,
If you really wanted to know how the pandemic affected this very specific population, you should better justify it in the introduction. I understand that it may have some importance from the point of view of the students (who saw face-to-face classes interrupted) or the teachers (who surely had to adapt to online teaching quickly) but none of that is mentioned; modifying the introduction in this direction is the best honorable solution.
On the other hand, the topic seems somewhat outdated. The data collection was carried out from September to November 2021, and we are in April 2024, meaning that the authors should somehow justify this striking gap, even if it is alleging that during these 3 years they carried out the analysis of the data collected.
Apart from that it was a pleasure to review this paper as the authors have shown a clarity and conciseness that should be appreciated; they present their results in a clearly written and well-organized without superfluous arguments neither unnecessary detours. The information provided is comprehensive and I like the way it is shown.
I have provided some comments with minor mistakes and methodological concerns that I found as follows:
In the line 2-Remove the word “title” from the title.
In the line 16- The name of the correspondence author should be the same as the author in the filiations (Heba M. Mamdouh)
In the line 17-In the abstract I would remove the boldface type in Background, Methodology, Results…, actually, I would remove these words directly.
In the line 32- “…The analysis showed that the pandemic has had a detrimental effect on this sample of university staff's levels of anxiety and depression…” the phrase is ambiguous, it is misunderstood, so, please, rewrite it.
In the line 34- the sentence “This study highlighted areas where the responsible authorities can intervene…” in uninformative so, please say at least one area or remove it.
In the line 37- remove “and” before resilience.
From the introduction till the end an extensive editing of English language is required.
In the line 174, in the ethical approval and consent statement the authors should specify which ethics review board approved this research.
The scales used to assess the mental health status of the sample seem correct.
In the line 300 I would add an extra hypothesis to explain the gender differences because both ones provided are quite similar.
In the line 350 Strengths and limitations remove the sentence: “It is impossible to dispute the uniqueness of 351 the data used in this study”
In the line 389 change: “…Continuous monitoring of the psychological status…” maybe you mean regular controls or something similar.
To end I would like to congratulate the authors for the numerous bibliographic references provided.
Kind regards
The reviewer
Comments on the Quality of English LanguageI understand the article but I feel that and extensive editing of English is needed to be done.
Author Response
We thank the editorial team and the reviewers for their careful evaluation of the manuscript, which has helped us tosubstantially improve our manuscript, titled “Understanding the Impact of COVID-19 on Mental Health among a sample of University Workers in the United Arab Emirates.” We have provided a point by point response for each suggestion and question raised by reviewer 2 in the text below. We hope our responses are clear and satisfactory.
Reviewer’s comment |
Feedback |
|
|
The topic seems somewhat outdated. The data collection was carried out from September to November 2021, and we are in April 2024, meaning that the authors should somehow justify this striking gap, even if it is alleging that during these 3 years they carried out the analysis of the data collected.
|
Kindly note that, we collected data from both the university students (larger sample) and university staff. We published the data for the university students in late 2022, for that it took us that time. However, topics related to COVID-19 are still researched heavily, particularly those topics that are related to the pandemic’ impact on the health status (including the mental health). Results of such studies could be used for shaping the health system needs in case of future outbreaks. |
Introduction |
|
If you really wanted to know how the pandemic affected this very specific population, you should better justify it in the introduction. I understand that it may have some importance from the point of view of the students (who saw face-to-face classes interrupted) or the teachers (who surely had to adapt to online teaching quickly) but none of that is mentioned; modifying the introduction in this direction is the best honorable solution. |
Response: As per the reviewer’s feedback, the introduction has been modified to include a full paragraph about the mental health impact of this particular population, as seen in lines 64-71.
|
In the line 2-Remove the word “title” from the title. |
Title word was removed |
In the line 16- The name of the correspondence author should be the same as the author in the filiations (Heba M. Mamdouh)
|
Was corrected accordingly. |
In the line 17-In the abstract I would remove the boldface type in Background, Methodology, Results…, actually, I would remove these words directly. |
The sections headings were removed as advised. |
In the line 32- “…The analysis showed that the pandemic has had a detrimental effect on this sample of university staff's levels of anxiety and depression…” the phrase is ambiguous, it is misunderstood, so, please, rewrite it. |
As advised by the reviewer, the sentence was rephrased as seen in lines 35, 36. |
In the line 34- the sentence “This study highlighted areas where the responsible authorities can intervene…” in uninformative so, please say at least one area or remove it.
|
As per the journal specification, the abstract is limited to 200 words. For that we cannot mention a more detailed policy statement. Accordingly, a sentence was added and detailed in the conclusion section as advised in lines 390 -394. |
In the line 37- remove “and” before Methods resilience. |
And was removed |
|
|
In the line 174, in the ethical approval and consent statement the authors should specify which ethics review board approved this research. |
The committee name was added in line 190. |
Discussion: |
|
In the line 350 Strengths and limitations remove the sentence: “It is impossible to dispute the uniqueness of 351 the data used in this study” |
The sentence was removed as advised. |
In the line 389 change: “…Continuous monitoring of the psychological status…” maybe you mean regular controls or something similar.
|
It has been changed to regular monitoring, as seen in line 412. |
Reviewer 3 Report
Comments and Suggestions for Authors
The title of the article, “Understanding the Impact of COVID-19 on Mental Health Among a Sample of University Staff in the United Arab Emirates,” is incorrect. COVID-19 is a disease caused by the SARS-CoV-2 virus. Only a small proportion of participants were diagnosed with Covid-19. It is known that in the course of this disease, in addition to systemic and respiratory symptoms, various neurological symptoms and mental health disorders may occur. The occurrence of mental health disorders in the remaining respondents was not related to COVID-19 disease, but to the COVID-19 epidemic and the risk of infection with the SARS-CoV-2 virus. Therefore, I suggest that, in addition to changing the title, people who have had COVID-19 should be excluded from the analyses.
The study itself is very interesting. It was carried out correctly using the appropriate scales to assess the mental health status.
Author Response
We thank the editorial team and the reviewers for their careful evaluation of the manuscript, which has helped us to substantially improve our manuscript, titled “Understanding the Impact of COVID-19 on Mental Health among a sample of University Workers in the United Arab Emirates.” We have provided a point by point response for each suggestion and question raised by reviewer 3 in the text below. We hope our responses are clear and satisfactory.
The study itself is very interesting. It was carried out correctly using the appropriate scales to assess the mental health status.
Reviewer’s comment |
Feedback |
Title |
|
The title of the article, “Understanding the Impact of COVID-19 on Mental Health Among a Sample of University Staff in the United Arab Emirates,” is incorrect. COVID-19 is a disease caused by the SARS-CoV-2 virus. |
Response: As advised by the respected reviewer, the term Pandemic was added to the title and that will fix the issues related to COVID-19 as a disease.
|
Methodology |
|
Only a small proportion of participants were diagnosed with Covid-19. It is known that in the course of this disease, in addition to systemic and respiratory symptoms, various neurological symptoms and mental health disorders may occur. The occurrence of mental health disorders in the remaining respondents was not related to COVID-19 disease, but to the COVID-19 epidemic and the risk of infection with the SARS-CoV-2 virus. Therefore, I suggest that, in addition to changing the title, people who have had COVID-19 should be excluded from the analyses.
|
Response: The study is concerned about assessing the potential effects of the pandemic on the mental health of this sample. In addition, comparing the mental health status (through using the 3 scales) of those who had COVID-19 to those who did not had been infected is a main aim of this research study. Data from tables 2, 3 show these comparisons and examine the effect of COVID-19 pandemic on both who were infected and the uninfected participants. As shown in Fig. 1. 46.1%of the participants mentioned they were diagnosed with COVID-19. However, |
Reviewer 4 Report
Comments and Suggestions for Authors
I would like to say thanks for the opportunity to review this article.
The article presented has an interesting and actual theme with relevance for the improvement of health interventions for improving mental health in university workers.
Overall, the article has a scientific and appropriate writing, including all the components of a good scientific research.
The title and abstract are related with the content. The keywords are linked to the research and majorly indexed. Abstract does not have a resume of introduction and literature review.
Introduction and literature review allow the framing of the theme and the research itself.
The main goal is well explained.
Methodology is scientifically appropriate. It is not clear if there were any inclusion and exclusion criteria in the sample. The impact of COVID-19 was assessed by 7 questions, but in the results (graphic) only four items appear. It is not clear the concept analysed (impact of..) and why it is considered an impact in those situations. It is also noticed that there is a big gap between the group impacted (85 % of the sample) and non-impacted (15% of the sample), which may influence results.
Results are adequate, but tables and text are not organized properly. Tables and graphs should immediately succeed their explanation text, and not come in the end all together, to be easier to understand and analyse results.
The article refers usually (including in the goal) to an analysis of the impact of COVID-19 in mental health of the sample. It also refers to the impact of sociodemographic variables on mental health. There was made an analysis of three variables (resilience, anxiety, and depression) during COVID-19, but there were no statistical analysis or method that allow to understand the causality. Results presented does not answer to the impact of COVID. It is a possible reason to be discussed, but not a certainty.
Discussion is done according to the results of the study, with the comparison with other study results and a good analysis.
Conclusion is good and adequate, and authors present limitations and implications that should be improved considering the sample selected.
References are pertinent, adequate, and majorly recent.
Proposals of improvement:
- Complete abstract
- Clarify methodology, namely in sample exclusion and inclusion criteria, and in impact of COVID-19 assessment
- Reorganize results
- Correct all the expressions about impact of COVID n mental health or other variables impact, because that was not tested
Thank you.
Author Response
We thank the editorial team and the reviewers for their careful evaluation of the manuscript, which has helped us to substantially improve our manuscript, titled “Understanding the Impact of COVID-19 on Mental Health among a sample of University Workers in the United Arab Emirates.” We have provided a point by point response for each suggestion and question raised by reviewer 4 in the text below. We hope our responses are clear and satisfactory.
Reviewer’s comment |
Feedback |
Abstract |
Response: Kindly note that, as per the journal’s requirements (and format), the abstract should be a single paragraph of 200 words. This extremely limits the addition of many basic important information to the abstract and the submission could be withheld if the number of words exceed the limit. |
Methods |
|
Methodology is scientifically appropriate. 1-The impact of COVID-19 was assessed by 7 questions, but in the results (graphic) only four items appear. It is not clear the concept analysed (impact of..) and why it is considered an impact in those situations. It is also noticed that there is a big gap between the group impacted (85 % of the sample) and non-impacted (15% of the sample), which may influence results.
2- Clarify in the methodology the sample exclusion and inclusion criteria, and in impact of COVID-19 assessment |
1-As suggested, the COVID_19 impact items were explained more to show how the 7 questions were re-grouped (as seen in lines 121- 129. Below are the detailed questions as per the questionnaire. 1. Have you been tested positive for Covid-19 and suffered from COVID-19 symptoms? 2. Has anyone in your family been tested positive for Covid-19? 3. Has anyone of your close friends been tested positive for Covid-19? 1,2, 3 were grouped together in the analysis. 4. Has anyone of your close friends died from complications of Covid-19? 5. Has anyone in your family died from complications of Covid-19? 4,5 were grouped together in the analysis.
6. Do you work in a setting where you witness people die from complications of Covid-19 ( Such as supervision or placement at hospitals, clinics or similar settings) ? 7. Do you work in a setting where you could be exposed to Covid-19 (e.g. Supervision or work-placement at hospitals, first responder, or other settings that require close human contacts)? 6,7 were grouped together in the analysis.
Directly impacted was meant for those who have tested positive and got the VOVID-19 symptoms. Indirectly impacted are for those who answered yes on the other questions. |
Results |
|
1. Results are adequate, but tables and text are not organized properly. Tables and graphs should immediately succeed their explanation text, and not come in the end all together, to be easier to understand and analyse results.
2. The article refers usually (including in the goal) to an analysis of the impact of COVID-19 in mental health of the sample. It also refers to the impact of sociodemographic variables on mental health. There was made an analysis of three variables (resilience, anxiety, and depression) during COVID-19, but there were no statistical analysis or method that allow to understand the causality. Results presented does not answer to the impact of COVID. It is a possible reason to be discussed, but not a certainty. - Correct all the expressions about impact of COVID on mental health or other variables impact, because that was not tested
|
Response: 1. True, tables and text are not organized properly, but this was done to follow the journal submission requirements. The results section will be re-organized by the journal editorial team once the manuscript accepted (as per their guidelines).
2. We actually ran some other tests to assess the impact COVID-19 on the socio-demographics. The Independent-sample t tests were used to test the difference of the mean scores for the three scales and the COVID-19 impact. Only for female participants the difference in mean scores for PHQ-9 and GAD-7 those impacted by COVID-19 and not impacted by COVID-19. We also tested for the interaction between the COVID-19 impact and other socio-demographics and the mean scores of PHQ 9, GAD-7, and CD-RISC 10 psychometric scales using the tests of Between-Subjects Effects and unfortunately all gave non-statistically significant results. Mostly due to the small sample size and the smaller sample of those who had a COVID-19 impact, a lot of variables gave insignificant results when we tested for significance. We can provide the data in appendixes, though no significant results were obtained (of no benefit). -The impact of COVID-19 in mental health of the sample was presented in the results under the following tables: Table 3. Independent sample T-test comparing the mean scores (± SD) of the PHQ- 9, GAD-7 and CD-RISC -10 psychometric scales by COVID-19 impact. Table 5. Two-way ANNOVA for comparing the differences in mean scores of PHQ 9, GAD-7 & and CD-RISC 10 scales among the participants by COVID-19 impact and gender. 2 ways ANOVA were used to also to test mean scores of PHQ 9, GAD-7 & and CD-RISC 10 scales among the participants by COVID-19 impact and other demographics (including the nationality, age group, marital status and the results were all non-significant).
|
Discussion |
|
Discussion is done according to the results of the study, with the comparison with other study results and a good analysis.
|
Response: Well noted thank you.
|
Conclusion |
|
Conclusion is good and adequate, and authors present limitations and implications that should be improved considering the sample selected.
|
Response: Well noted thank you.
|
Round 2
Reviewer 1 Report
Comments and Suggestions for Authors
Dear authors
Thank you for addressing the comments I have made. However, below are a few previous comments that have yet to be addressed or can be addressed better.
Methodology
1. The comment on the study population is not being addressed, although the minimum number has been informed. Authors can also inform what type of sampling is being used.
Results
1. I suggest the authors state in the manuscript that a higher impact analysis has been done and academically justify the results as a summary per my previous comment in the results section. I appreciate that the authors have taken into consideration other statistical analyses, and insignificant results are still results for your case study.
Thank you.
Author Response
Feedback Reviewer 1:
We thank the editorial team and the reviewers for their careful evaluation of the manuscript, which has helped us to substantially improve our manuscript, titled “Understanding the Impact of COVID-19 on Mental Health among a sample of University Workers in the United Arab Emirates.” We have provided a point-by-point response for each suggestion and question raised by reviewer 1 in the text below. We hope our responses are clear and satisfactory.
Review:
Methodology
- The comment on the study population is not being addressed, although the minimum number has been informed. Authors can also inform what type of sampling is being used.
Response:
The type of sampling was added as advised in lines 100, and 101.
An inconvenience sampling technique was followed in this study in which the selection of participants from the target population was based on ease of access.
Results
- I suggest the authors state in the manuscript that a higher impact analysis has been done and academically justify the results as a summary per my previous comment in the results section. I appreciate that the authors have taken into consideration other statistical analyses, and insignificant results are still results for your case study.
Response:
As per the reviewer’s feedback, the sentence was added in lines 261-265.
We tested for the interaction between the COVID-19 impact variable and other socio-demographics and the mean scores of PHQ 9, GAD-7, and CD-RISC 10 psychometric scales using the tests of Between-Subjects Effects and unfortunately, the results were non-statistically significant (data is provided as a supplementary material).

Reviewer 3 Report
Comments and Suggestions for Authors
The authors have made corrections to the text of the article, which I fully accept. I suggest publishing the article in present form.
Author Response
We thank the editorial team and the reviewers for their careful evaluation of the manuscript, which has helped us to substantially improve our manuscript.
Reviewer 4 Report
Comments and Suggestions for Authors
All sugestions were considered.
Author Response
We thank the editorial team and the reviewers for their careful evaluation of the manuscript, which has helped us to substantially improve our manuscript